# Cu(II)-Loaded Polydopamine-Coated Urchin-like Titanate Microspheres as a High-Performance IMAC Adsorbent for Hemoglobin Separation

**DOI:** 10.3390/molecules29071656

**Published:** 2024-04-07

**Authors:** Qian Zhang, Linlin Hu, Jianyu Yang, Pengfei Guo, Jinhong Wang, Weifen Zhang

**Affiliations:** 1School of Pharmacy, Shandong Second Medical University, Weifang 261053, China; zhang_qian0424@163.com (Q.Z.); hull@sdsmu.edu.cn (L.H.); zhangwf@sdsmu.edu.cn (W.Z.); 2Shandong Engineering Research Center for Smart Materials and Regenerative Medicine, Weifang 261053, China; 3School of Materials Science and Engineering, Suzhou University of Science and Technology, Suzhou 215009, China; jianyuyang@usts.edu.cn

**Keywords:** urchin-like structure, titanate, adsorbent, separation, hemoglobin

## Abstract

Immobilized metal ion affinity chromatography (IMAC) adsorbents generally have excellent affinity for histidine-rich proteins. However, the leaching of metal ions from the adsorbent usually affects its adsorption performance, which greatly affects the reusable performance of the adsorbent, resulting in many limitations in practical applications. Herein, a novel IMAC adsorbent, i.e., Cu(II)-loaded polydopamine-coated urchin-like titanate microspheres (Cu-PDA-UTMS), was prepared via metal coordination to make Cu ions uniformly decorate polydopamine-coated titanate microspheres. The as-synthesized microspheres exhibit an urchin-like structure, providing more binding sites for hemoglobin. Cu-PDA-UTMS exhibit favorable selectivity for hemoglobin adsorption and have a desirable adsorption capacity towards hemoglobin up to 2704.6 mg g^−1^. Using 0.1% CTAB as eluent, the adsorbed hemoglobin was easily eluted with a recovery rate of 86.8%. In addition, Cu-PDA-UTMS shows good reusability up to six cycles. In the end, the adsorption properties by Cu-PDA-UTMS towards hemoglobin from human blood samples were analyzed by SDS-PAGE. The results showed that Cu-PDA-UTMS are a high-performance IMAC adsorbent for hemoglobin separation, which provides a new method for the effective separation and purification of hemoglobin from complex biological samples.

## 1. Introduction

Protein research is of great significance for exploring the pathogenesis of diseases, searching for disease-related molecular markers, developing drug targets, and preventing and treating major human diseases [1,2,3,4]. Protein research is inseparable from protein separation and purification [5,6,7]. Hemoglobin plays a vital role in numerous physiological activities, including oxygen binding and transport in the blood and regulation of blood acid-base balance [8,9]. Abnormal levels of hemoglobin could predict the occurrence of diseases, such as anemia, blood clots, cancer, and kidney failure. However, hemoglobin exists mostly in the complex biological microenvironment, along with interfering substances such as albumin, peptides, and small molecules [10,11]. Moreover, protein biomarkers can be used as new molecular targets for the design of new drugs, and the high content of hemoglobin in blood samples may interfere with the selective detection and isolation of low-abundance protein [12].Therefore, the separation and enrichment of hemoglobin from complex biological samples is critical for new drug research and the detection of some diseases. Among various separation technologies, adsorbents are preferred because of their low cost and simple operation. Although many adsorbents have been developed in the past, they all have shortcomings such as low adsorption capacity, poor stability, and the fact that they cannot be recycled, which greatly hinder their practical application. Therefore, it is imperative to develop new adsorbents with high adsorption capacity and good selectivity for hemoglobin.

Over the past decade, various separation techniques for hemoglobin have been developed, e.g., molecularly imprinted polymers (MIPs), immobilized metal affinity chromatography (IMAC), metal oxide affinity chromatography (MOAC) [13,14,15]. Many separation techniques used in the process of protein separation have the problems of low efficiency of protein desorption and difficult synthesis of adsorbent [16]. The affinity interactions between metal cations on IMAC and specific amino acids in proteins [17], simple synthesis, and high adsorption elution efficiency enable widespread application of IMAC for selective isolation and purification of specific proteins. For example, Du et al. synthesized an immobilized Cu^2+^ adsorbent (Cu^2+^PCM-15) for the selective adsorption of bovine hemoglobin [18]. Ding and colleagues synthesized copper (II) affinity-DTPA functionalized magnetic composite (Fe_3_O_4_@SiO_2_@NH_2_@DTPA@Cu_2_O_3_) for the purification of hemoglobin. The results show that the separation efficiency for hemoglobin of the as-synthesized adsorbent greatly improved by the loading of copper ions [19]. Jiang et al. prepared the yolk shell Fe_3_O_4_@NiSiO_3_ microspheres by a simple hydrothermal method. Due to the large specific surface area, it can load a large amount of Ni NPs and has a high adsorption capacity for hemoglobin [20]. Among the transition metal ions, copper ions show the strongest affinity for proteins rich in histidine residues, because copper ions not only bind specifically to the histidine site but they also interact strongly with the C-terminus of the peptide [21,22]. The transition metal decoration on a variety of carrier materials with large surface area is beneficial for improving the adsorption affinity sites and enhancing the adsorption capacity. At present, various shapes of materials for use as adsorbents, such as spherical, sheet, rod, etc., are used in the field of hemoglobin separation and purification [23,24,25]. In these materials, urchin-like microspheres used as adsorbents have more binding sites to target proteins than spherical materials. A variety of titanate-based adsorbents have been gradually developed and applied to the enrichment of biological macromolecules because of their excellent physical and chemical properties, excellent structural stability, and good biocompatibility [26,27]. 

In this study, novel Cu (II)-loaded polydopamine-coated urchin-like titanate microspheres (Cu-PDA-UTMS) were prepared via the coating of polydopamine and chelating of copper ions onto the surface of titanate microspheres. The as-synthesized urchin-like microspheres are used as an IMAC adsorbent by virtue of the specific affinity interaction between the copper ions on the surface of the microspheres and the target protein. Because copper ions can be dispersed onto urchin-like dopamine-coated titanate microspheres, additional affinity sites are exposed to bind more hemoglobin. In addition, Cu-PDA-UTMS is stable when used in the adsorption−desorption process, and there is no need for re-incubation with metal salts to maintain its adsorption performance for continued use. The adsorption of hemoglobin with Cu-PDA-UTMS from human whole blood further demonstrates the application potential in complex samples.

## 2. Results and Discussion 

### 2.1. Synthesis and Characterization

Figure 1 shows the synthetic route of Cu (II)-loaded polydopamine-coated urchin-like titanate microspheres (Cu-PDA-UTMS) and the adsorption/desorption procedure for hemoglobin by using Cu-PDA-UTMS as an IMAC adsorbent. In Figure 1a, the synthesis process of Cu-PDA-UTMS can be divided into five steps. In the first step, Na-titanate nanotubes (Na-TNT) are prepared by hydrothermal reaction of titanium dioxide with NaOH. Urchin-like Na-titanate microspheres (Na-UTMS) are synthesized by hydrothermal reaction of Na-TNT with H_2_O_2_ and NaOH in the second step. In the third step, urchin-like protonated titanate microspheres (H-UTMS) are obtained after Na-UTMS is treated with HNO_3_. Polydopamine is coated on the surface of titanate microspheres by self-polymerization in the fourth step. In the last part, copper ions are modified on the surface of PDA-UTMS through metal coordination to obtain the final product (Cu-PDA-UTMS). In the process that follows, Cu-PDA-UTMS is used as an IMAC adsorbent to isolate and purify hemoglobin. This separation and purification process utilizes the specific affinity between hemoglobin and copper ions on the surface of the microsphere to adsorb hemoglobin from the complex sample, and it uses the competitive interaction between the eluent (CTAB) and the adsorbent towards hemoglobin to elute the hemoglobin trapped on the adsorbent. Figure 1b shows the separation and purification process of hemoglobin. Specifically, the adsorbent is first used to selectively adsorb hemoglobin from complex samples. Then, the adsorbent bound with hemoglobin is separated from the interfering protein by centrifugation. Finally, the hemoglobin captured on the adsorbent is eluted for later analysis.

The morphologies of PDA-UTMS and Cu-PDA-UTMS were firstly characterized by transmission electron microscopy (TEM). As seen in Figure 1a,b, the polydopamine-coated titanate microspheres show a good sea urchin-like shape formed by the clustering together of countless tubular structures, which provide a rich specific surface area for binding more copper ions. As displayed in Figure 1c,d, after the coating of copper ions onto the polydopamine-coated titanate microspheres, the material still exhibits a good sea urchin-like shape, which provides a rich specific surface area for binding more target proteins. In addition, in the TEM element mapping of Cu-PDA-UTMS (Appendix A), C, Cu, N, O, and Ti are uniformly distributed on titanate microspheres, respectively, which proves the success of polydopamine encapsulation and the coating of copper ions onto titanate microspheres. 

A scanning electron microscope (SEM) method was then adopted to characterize the morphologies of different products, including Na-TNT, Na-UTMS, H-UTMS, PDA-UTMS, and Cu-PDA-UTMS, respectively. From Appendix A, it can be seen that commercial spherical titanium dioxide with an average particle size of 25 nm successfully synthesized disorderly stacked Na-titanate nanotubes (Na-TNT). As shown in Appendix A, the disorderly stacked Na-titanate nanotubes were formed into sea urchin-like Na-titanate microspheres (Na-UTMS) with ordered aggregation of nanotubes through the hydrothermal reaction. There is no significant shape change in the urchin-like protonated titanate microspheres (H-UTMS) obtained by nitric acid treatment in Appendix A. From Figure 2a,b, the surface of the urchin-like titanate microspheres becomes rough after they are coated with polydopamine. After decorating copper ions onto polydopamine on the surface of the microsphere through coordination, Figure 2c,d shows that Cu-PDA-UTMS have similar urchin-like morphology compared with PDA-UTMS, indicating the successful preparation of IMAC adsorbent with urchin-like morphology.

The elemental content of created microspheres, including H-UTMS, PDA-UTMS and Cu-PDA-UTMS, was analyzed by SEM-EDS analyzer. From Appendix A, the decreases in Ti content from 35.04% to 19.98%, O content from 64.96% to 48.19%, and the occurrence of a C content of 31.83% in PDA-UTMS indicate the successful encapsulation of polydopamine on H-UTMS. The presence of 1.51% copper content in Cu-PDA-UTMS demonstrates the successful modification of copper ions on PDA-UTMS. From Appendix A, the success of dopamine coating and copper ion loading can also be demonstrated by the change in color of H-UTMS, PDA-UTMS, and Cu-PDA-UTMS from white to light brown to dark brown.

The chemical composition of Cu-PDA-UTMS was further determined by XPS analysis. As shown in the Figure 3a, the five peaks of the full spectrum of Cu-PDA-UTMS are at the binding energies of 934, 530, 459, 401, and 285 eV, corresponding to Cu 2p, O 1s, Ti 2p, N 1s, and C 1s, respectively. This indicates the successful preparation of Cu (II)-loaded polydopamine-coated urchin-like titanate microspheres. In the high resolution Cu 2p spectrum shown in Figure 3b, the binding energies at 952.9, 942.4, and 933.0 eV correspond to the peak of Cu 2p_1/2_, the satellite peak of Cu 2p_3/2_, and the main peak of Cu 2p_3/2_, respectively, which clearly proves the presence of the characteristic feature of the Cu^2+^ species [25]. 

The wide-angle X-ray diffraction pattern (XRD) of Na-TNT, Na-UTMS, H-UTMS PDA-UTMS, and Cu-PDA-UTMS were displayed in Figure 3c. For Na-TNT, the diffraction peaks at 2θ = 7.7°, 24.3°, 28.2°, and 48.2° could be assigned to the (200), (110), (310), and (020) planes of Na-titanate nanotubes [28,29]. For Na-UTMS and H-UTMS, the diffraction peaks were well matched with the structure of orthorhombic NaxH_2_-xTi_2_O_5_·H_2_O and H_2_Ti_2_O_5_·H_2_O, respectively [30]. For PDA-UTMS and Cu-PDA-UTMS, the diffraction peaks were similar to those of H-UTMS, which proves that the polydopamine encapsulation and copper ion modification on the microspheres have no significant effect on the structure of titanate. The N_2_ adsorption/desorption isotherm of Cu-PDA-UTMS showed a specific surface area of 97.64 m^2^ g^−1^ from BET analysis. The isotherm (Figure 3d) shows a typical type IV isotherm with a type H3 hysteresis loop, indicating the presence of a porous structure consisting of aggregates of nanotubes forming slit-shaped pores [31]. This observation is consistent with those derived from the TEM images.

### 2.2. Protein Adsorption Behaviors 

Since the affinity behavior of adsorbents towards proteins varies greatly in different pH environments, we investigated the adsorption properties towards proteins by changing the pH value of the adsorption environment. The adsorption efficiencies for Hb and BSA by Cu-PDA-UTMS and PDA-UTMS at different pH values are shown in Figure 4a,b. As can be seen from Figure 4a, the adsorption efficiencies for the protein by PDA-UTMS reached the maximum at pH value close to the isoelectric point. That is to say, when pH is 7.0, the adsorption efficiency of Hb is the highest, and when pH is 5.0, the adsorption efficiency of BSA is the highest. At isoelectric points, proteins are neutrally charged and exhibit greater hydrophobicity. At this time, the functional groups of polydopamine covered on the titanate microspheres, such as hydroxyl groups, amino groups, benzene rings, nitrogen heterocyclic rings, etc., can produce strong hydrophobic interactions and hydrogen bonding with hemoglobin. From Figure 4b, the relationship between pH and for Hb and BSA by Cu-PDA-UTMS is similar to that between pH and the adsorption efficiencies for Hb and BSA by PDA-UTMS. However, the adsorption efficiency of Cu-PDA-UTMS for hemoglobin is obviously higher than that of PDA-UTMS for hemoglobin. This is due to the formation of a strong metal affinity between Cu-PDA-UTMS and hemoglobin. Hemoglobin has 24 exposed histidine residues, while bovine serum albumin has only 2 exposed histidine residues [32]. Therefore, the adsorption efficiency for Hb by Cu-PDA-UTMS at the optimal pH is significantly higher than that for BSA at this pH. Therefore, it is preferred to perform adsorption experiments for Hb under the condition of pH 7.

In order to better evaluate the adsorption selectivity by Cu-PDA-UTMS, several proteins, including β-casein (β-cas), bovine serum albumin (BSA), immunoglobulin G (IgG), γ-globulin (γ-glo), and hemoglobin (Hb) were tested. A total of 0.1 mg Cu-PDA-UTMS was applied to adsorb these proteins (1.0 mL, 100 mg L^−1^, pH 7). From Figure 4c, the adsorption efficiencies of Cu-PDA-UTMS for β-cas, BSA, IgG, γ-glo, and Hb were 6.0%, 18.8%, 40.4%, 50.4%, and 92.8%, respectively. These results showed that the adsorption efficiency for Hb by Cu-PDA-UTMS was better than that for other proteins, which was attributed to the strong metal affinity between hemoglobin and copper ions on the surface of the adsorbent. 

In practical application, the effect of ionic strength on protein adsorption should be considered. We investigated the adsorption behaviors towards proteins by Cu-PDA-UTMS under different concentrations of NaCl. As shown in Figure 4d, over a wide range of ionic strengths, the effect of ionic strength on adsorption efficiency of Cu-PDA-UTMS for Hb and BSA is negligible. These results clearly show that the electrostatic interaction is not involved in the adsorption of hemoglobin by Cu-PDA-UTMS. This offers great potential for the selective separation of hemoglobin from complex biological sample matrices at high ionic strength.

Measurement of the adsorption isotherm for hemoglobin by Cu-PDA-UTMS was carried out by adsorption of hemoglobin in the concentration range of 0.1 to 0.7 mg mL^−1^ by 0.1 mg Cu-PDA-UTMS at room temperature. As shown in Figure 5a, Cu-PDA-UTMS provided an adsorption capacity of 2704.6 mg g^−1^ for hemoglobin. Table 1 summarizes the adsorption capacity of different adsorbents for hemoglobin. It can be seen that the adsorption capacity of Cu-PDA-UTMS for Hb is significantly higher than that of other adsorbents. This is due to the fact that Cu-PDA-UTMS have more metal affinity sites for hemoglobin by virtue of its special sea urchin-like structure.

For subsequent biological studies, it is very necessary to elute the adsorbed Hb from Cu-PDA-UTMS. Therefore, elution experiments were carried out with a series of potential stripping reagents, i.e., 0.1% CTAB, 0.5% SDS, 0.1 M Na_2_CO_3_, 0.1 M acetic acid, and 0.1 M imidazole. As demonstrated in Figure 5b, the results showed that the adsorbed Hb was easily eluted with 0.1% CTAB with an elution rate of 86.8%, which is attributed to the competitive interaction between CTAB and the adsorbent that enables elution of the hemoglobin trapped on the adsorbent (Figure 1a). In addition, by comparing the circular dichroism (CD) of standard hemoglobin and hemoglobin solution after adsorption by the Cu-PDA-UTMS and recovery in 0.1% CTAB solution, it can be found that the α-helix structure is almost the same, which shows the conformational changes of hemoglobin were essentially not changed (Appendix A).

The reusability of Cu-PDA-UTMS as an adsorbent was investigated by the continuous adsorption/desorption of hemoglobin. Specifically, Cu-PDA-UTMS is used to repeat this process by adsorbing hemoglobin in 0.04 mol L^−1^ BR buffer (pH 7), desorbing hemoglobin by using 0.1% CTAB, and then continuing to adsorb hemoglobin. As shown in Figure 5c, the adsorption efficiency for Hb by Cu-PDA-UTMS did not change significantly after six consecutive adsorption/desorption processes. The results clearly demonstrate the applicability of Cu-PDA-UTMS as a reusable adsorbent for Hb separation.

The study of the protein’s adsorption behavior under different pH conditions, salt concentrations, and so on has proven its good potential for use. Next, the practicability of Cu-PDA-UTMS in the selective adsorption of hemoglobin was tested by using diluted human whole blood without other processing as a biological sample. Specifically, 0.1 mg Cu-PDA-UTMS was used to enrich 100-fold dilutions human whole blood samples diluted in 0.04 mol L^−1^ BR buffer (pH 7), after which the protein was bound by the adsorbent and eluted by 0.1% CTAB. The adsorption and elution of proteins were analyzed by SDS-PAGE with separation gel (12%) under 80 V and laminated gel (5%) under 120 V. As shown in the SDS-PAGE results (Figure 5d), these bands of the 100-fold diluted human whole blood sample (lane 2) contain mainly serum albumin (66.4 kDa) and α or β globin chains of Hb (14.3 kDa). After the elution of adsorbed proteins from Cu-PDA-UTMS, lane 4 showed that the supernatant mostly contained the band of corresponding α or β globin chains of Hb at 14.3 kDa. As a comparison, the band for the 0.1 mg mL^−1^ hemoglobin standard was also shown (lane 5). From SDS-PAGE analysis, it can be seen that the proportion of hemoglobin enriched from blood is significantly higher than that of human whole blood samples before enrichment. The results further indicated that Cu-PDA-UTMS displayed good selectivity for Hb and could be a promising candidate for selective separation of Hb from complicated biological samples. As we know, the hemoglobin purification/separation is a multi-stage process where the number of purification steps depends on application of the isolated hemoglobin (diagnostic reagent, food additive, cell culture additive, or therapeutic agent or it just needs to be removed due to interference with the molecules to be analyzed). In light of our results, this new adsorber, with proven good characteristics, can be used as one step in the complex hemoglobin purification process.

## 3. Materials and Methods

### 3.1. Chemicals and Reagents

Hemoglobin from bovine blood (Hb), bovine serum albumin (BSA), Immunoglobulin G from human serum (IgG), γ-globulin from bovine serum (γ-glo), and β-casein from bovine milk (β-cas) were obtained from Macklin Biochemical Co., Ltd. (Shanghai, China) or Sigma-Aldrich (St. Louis, MO, USA). Titanium oxide (P25, 99.5%) was supplied by Aladdin (Shanghai, China). Sodium chloride (NaCl, 99%, AR), sodium hydroxide (NaOH, 99%, AR), hydrochloric acid (HCl), acetic acid (HAc), sodium carbonate (Na_2_CO_3_), sodium dodecyl sulfonate (SDS), tris (chydroxymethyl) aminomethane (Tris), hexadecyl trimethyl ammonium bromide (CTAB, 99%), copper (II) sulfate pentahydrate (CuSO_4_·5H_2_O, 99%), dopamine hydrochloride (98%), imidazole, boric acid (H_3_BO_3_), and phosphoric acid (H_3_PO_4_) were purchased from Sinopharm Chemical Reagent Co. or Macklin Biochemical Co., Ltd. (Shanghai, China). The ultra-pure water was purified by a Millipore system (Burlington, MA, USA). Premixed Protein Maker and Loading Buffer were purchased from Takara Bio Inc. (Nojihigashi, Japan). All reagents are used directly and without further purification. Human whole blood was provided by a healthy volunteer from Shandong Second Medical University. All experiments were performed in compliance with the Helsinki Declaration of 1975 (revised in 2008), as approved by the institutional committee of Shandong Second Medical University on human experiments and the Ethics Committee of the Academic Advisory Board, Shandong Second Medical University, China. Informed consent was obtained from all subjects for being included in this study. 

### 3.2. Instruments

Transmission electron microscopy (TEM) images and TEM element mapping were characterized on a JEM-2100F (JEOL, Akishima, Japan). Scanning electron microscope (SEM) images and scanning electron microscopy in combination with energy-dispersive X-ray spectrometry (SEM/EDS) analysis were made by using scanning electron microscopy (SU-8010, Hitachi, Tokyo, Japan). An X-ray diffractometer (XRD) with Cu/Co radiation from D8 ADVANCE Germany (Bruker, Ettlingen, Germany) was used to identify the crystalline phase. X-ray photoelectron spectroscopy (XPS) was recorded by using a Thermo Scientific ESCALAB 250XI X-ray spectrometer (Waltham, MA, USA). The N_2_ adsorption/desorption experiment was measured with a Micromeritics ASAP 2460 analyzer (Norcross, GA, USA). A UV–vis spectrophotometer (T6 New Century, Beijing, China) was used for the detection of protein concentration, and a vortex mixer (VM-0003M, Labyeah, Shanghai, China) was used for the incubation of protein. Separation of the actual protein sample was performed by a sodium dodecyl sulfate-polyacrylamide gel (SDS-PAGE) electrophoresis apparatus (Bio-Rad, Hercules, CA, USA).

### 3.3. Synthesis

#### 3.3.1. Fabrication of Na-Titanate Nanotubes (Na-TNT)

Na-TNT were prepared based on a hydrothermal reaction [39]. Briefly, 2.0 g of P25 was reacted with 40 mL of 10 mol L^−1^ NaOH solution in a Teflon-lined stainless steel autoclave at 120 °C for 24 h. The resulting product was washed several times with deionized water and dried at 60 °C to obtain Na-TNT.

#### 3.3.2. Fabrication of Urchin-like Na-Titanate Microspheres (Na-UTMS)

Na-UTMS were prepared via hydrothermal reaction with 0.2 g of Na-TNT, 40 mL of 1 mol L^−1^ NaOH solution, and 1.5 mL of H_2_O_2_ (30%) in a Teflon-lined stainless steel autoclave at 150 °C for 12 h [40]. The product was washed several times with deionized water and dried at 60 °C to obtain Na-UTMS.

#### 3.3.3. Fabrication of Urchin-like Protonated Titanate Microspheres (H-UTMS)

H-UTMS were obtained by dispersing 200 mg Na-UTMS powders in HNO_3_ aqueous solution (1 mol L^−1^) with violent stirring at room temperature for 24 h. The HNO_3_ solution was replaced every 12 h. Then, the solid product was washed several times with deionized water and dried at 60 °C.

#### 3.3.4. Fabrication of Polydopamine-Coated Urchin-like Titanate Microspheres (PDA-UTMS)

PDA-UTMS were obtained through the self-polymerization of dopamine on the surface of H-UTMS. Specifically, 5 mL of dopamine hydrochloride aqueous solution (1 mg mL^−1^) was slowly added to 10 mL of Tris (0.1 mg mL^−1^) aqueous solution with 50 mg of H-UTMS then stirred at room temperature for two hours, and the precipitation was collected through centrifugation. The final product was washed several times with deionized water and dried at 60 °C.

#### 3.3.5. Fabrication of Cu (II)-Loaded Polydopamine-Coated Urchin-like Titanate Microspheres (Cu-PDA-UTMS)

Cu-PDA-UTMS were prepared by a chelating reaction between copper ions and polydopamine on the PDA-UTMS. Specifically, 50 mg PDA-UTMS and 100 mL 0.1 mol L^−1^ CuSO_4_·5H_2_O aqueous solution were violently stirred at room temperature for 12 h. The synthesized product was separated by centrifugation and washed several times with deionized water followed by drying at 60 °C.

### 3.4. Protein Adsorption Behavior

Typically, the adsorbent was ultrasonically dispersed in aqueous medium to prepare 5 mg mL^−1^ suspension liquid. 20 μL suspension, i.e., 0.1 mg adsorbent, was used to adsorb the protein (100 mg L^−1^). The mixture (1 mL) was incubated by shaking vigorously for 60 min at room temperature to facilitate the adsorption of protein species. The adsorbent that captured the protein was eluted by a series of eluents at room temperature for 60 min. The aqueous medium at different pH values is regulated by a 0.04 mol L^−1^ Britton-Robinson (BR) buffer. The adsorption performance of the adsorbents was determined with UV–vis spectrophotometer according to the absorbance at 405 nm for hemoglobin of the supernatant before and after the adsorption of protein by the adsorbent and the absorbance of the eluent before and after the elution of protein. 

Equations related to the calculation of adsorption performance of the adsorbents. 

Protein adsorption capacity (*Q*, mg g^−1^) and adsorption efficiency (*η*, %) of the adsorbent are calculated according to Formulas (1) and (2). The recovery rate (%) of the extracted liquid was calculated according to Formula (3).
(1)$$Q=\frac{(C_{0}-C_{e})V}{m}$$
(2)$$\eta =\frac{\left(C_{0}-C_{e}\right)}{C_{0}}\times 100\%$$
(3)$$\mathrm{Recovery}=\frac{C_{L}}{C_{0}-C_{e}}\times 100\%$$
where *C*_0_ and *C_e_* (mg mL^−1^) represent the initial and equilibrium protein concentrations, respectively. The volume of protein solution and the weight of the adsorbent are represented by *V* (mL) and *m* (g), respectively. *C_L_* (mg mL^−1^) was the protein concentration in the eluent.

## 4. Conclusions

In summary, Cu(II)-loaded polydopamine-coated urchin-like titanate microspheres were constructed by loading Cu^2+^ onto the surface of polydopamine-coated urchin-like titanate microspheres. A series of characterization methods such as SEM, TEM, FT-IR, XRD, XPS, and N_2_ adsorption/desorption isotherms and so on demonstrated the successful preparation of Cu-PDA-UTMS. The as-prepared IMAC adsorbent can provide abundant metal affinity sites for binding hemoglobin. Cu-PDA-UTMS exhibit favorable adsorption selectivity for hemoglobin and the maximum adsorption capacity of Cu-PDA-UTMS for Hb was 2704.6 mg g^−1^, which was better than those of many other adsorbents. In addition, the reusable performance of Cu-PDA-UTMS was sustained for six cycles. Through SDS-PAGE analysis, Cu-PDA-UTMS has been shown to have successfully isolated hemoglobin from human whole blood samples, demonstrating its great potential for practical biological applications.

## Data Availability

Data will be made available on request.

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
