# Peer review of "Cu(II)-Loaded Polydopamine-Coated Urchin-like Titanate Microspheres as a High-Performance IMAC Adsorbent for Hemoglobin Separation"

_molecules, 2024, doi:10.3390/molecules29071656_

Round 1

Reviewer 1 Report

Comments and Suggestions for Authors

In the article “Cu(II)-loaded polydopamine-coated urchin-like titanate micro-spheres as a high-performance IMAC adsorbent for hemoglobin separation” the authors first showed physicochemical characteristics of new IMAC adsorber Cu-PDA-UTMS, and then describe its hemoglobin binding properties. The authors stated that adsorption capacity towards hemoglobin is up to 2.7046 g/L and that 0.1% CTAB elutes 86.8% of adsorbed hemoglobin. They also showed that the pH but not ionic strength influence hemoglobin adsorption, and the elution efficiency depend on stripping reagents. The authors also claimed that Cu-PDA-UTMS shows good reusability up to 6 cycles. However, the results also showed that Cu-PDA-UTMS is not a selective adsorber, specific only for hemoglobin. They adsorbed other proteins, and adsorption efficacy for IgG and BSA is approximately 50% and 20%. The elution efficiency for BSA and IgG is not reported.

General comments

Although hemoglobin binding properties of Cu-PDA-UTMS adsorber are promising, I think that the adsorbing characteristics of this matrices have to be studied more profoundly.

I suggest rejection or serious revision which should include new experiments.

Specific comments

Introduction

In Introduction section first paragraph should be deleted and replaced with new one. The sentence “Therefore, the separation and enrichment of hemoglobin from complex biological samples is critical for the treatment and diagnosis of some diseases.” is not clear. Modern diagnostic methods for detecting different hemoglobinopathies do not require neither purified hemoglobin nor large amounts of hemoglobin. Also, is not clear what kind of disease treatments the authors are referring to.

For thar reason I suggest that authors in the first paragraph of Introduction section report the data, existing in scientific literature, on potential biomedical and biotechnological application of purified or partially purified hemoglobin. In the light of existing data, the authors should more precisely explain the aim of this study and to assume potential application of Cu-PDA-UTMS adsorber.

Experimental section

The Molecules journal articles have “Materials & Methods” but not “Experimental section”. In addition, this section must be placed after Result and Discussion section. The authors should rearrange the manuscript according the journal demands.

Subsection 2.1. Chemicals and reagents

The last sentence in subsection “Human whole blood was provided by a healthy volunteer from Shandong Second Medical University” should be followed by the statement which is now in supplementary material “All experiments were performed in compliance with the Helsinki Declaration of 1975 (revised in 2008), as approved by the institutional committee of Shandong Second Medical University on human experiments, and approved by the Ethics Committee of Academic Advisory Board, Shandong Second Medical University, China. Informed consent was obtained from all subjects for being included in this

Study”.

In addition, Molecules journal demands that a statement including the project identification code, date of approval, and name of the ethics committee or institutional review board be stated in Section ‘Institutional Review Board Statement’ at the end the article. Therefore, I ask the authors to include this information (first of all, the number and date of issuing the ethical approval) in the “Institutional Review Board Statement”.

Subsection 2.4. Protein adsorption behavior

The detailed description of protein adsorption behavior must be given in the main text and not in Supplementary material.

The authors wrote that they mixed 0.1 mg adsorbent and 0.1 mg protein (if the protein concentration was 100 mg/L) in total volume of 1 ml, and then they incubated the suspension for 60 min. However, it is not clear if the adsorbent is washed after this incubation period (to remove unbound proteins and an excess of binding buffer) and if the eluent volume was also 1 ml.

If it's not a case, mass of proteins in starting solution and in the eluent should be calculated and “C” i.e., “protein concentration” should be replaced with the “protein mass” in Formula 1, 2 and 3.

Also, the authors should explain in detail how adsorption performance of the adsorbents are determined with UV-vis spectrophotometer.

Additional remark: The authors did not give the composition of binding buffer(s). They just stated the buffer pH value.

Results and Discussion

Major comment: The obtained results were not discussed at all, neither in the light of the results of other research groups studying IMAC nor in the light of possible application.

Subsection 3.2. Protein adsorption behaviors

1.      Although it is known that any IMAC adsorbent is not selective for a single protein species, and that, because of this properties, Cu-PDA-UTMS also binds other proteins besides hemoglobin, the authors demonstrated preferential binding of hemoglobin only by SDS PAGE of whole blood.

2.      The authors did not explain how they papered whole blood (which normally contains plasma proteins and intact erythrocytes, leukocytes and platelets) to be adsorbed to Cu-PDA-UTMS. If they added whole blood to adsorber, and then isolated hemoglobin, this means that adsorber provoked erythrocyte destruction (lysis) and release of free hemoglobin which than became available for the binding to adsorber.

If this is a case, the author should analyze the binding of erythrocyte cytoskeletal proteins (which can be regarded as unwanted contaminates in some of hemoglobin preparation) to Cu-PDA-UTMS.

3.      The author did not give the characteristics of SDS PAGE. They only reported the name of electrophoresis devices) but no other important data: method for preparing the samples for SDS PAGE, used sample buffer, gel thickens and percentage, protein load, name and manufacturer of protein makers. From my point of view the PAGE is not well performed and bands are not clear. Also, the authors should harmonize the marking of gel lines in the figure (lanes labeled with number) and in the text (lines labeled with letters).

4.      Molecular mass of hemoglobin is not 14.3 kD but 64.5 kDa. The mass of 14.3 kDa correspond to mass of alpha or beta globin chains. During SDS PAGE hemoglobin dissociate to their monomers, i.e., alpha and beta globin chains.

5.      The authors did not analyze if eluted hemoglobin is in its native Fe2+ tetramer conformation or denatured as a consequence of interaction with matrices. To verified it, more method is necessary to be applied and SAD-PAGE is not valid for such analysis. Without data on conformational stability of isolated hemoglobin the reported results can be regarded as preliminary.

6.      The authors did not analyze if lipids or lipoproteins (from whole blood i.e., from plasma and erythrocyte membrane) nonspecifically interacted with Cu-PDA-UTMS.

7.      The authors used whole blood which normally contains leucocytes. Leukocytes contain nucleic acids which can be released during the procedure and contaminate the isolated hemoglobin. This means that DNA and RNA binding to Cu-PDA-UTMS should be analyzed.

Conclusions

I think that statement “Cu-PDA-UTMS has successfully isolated hemoglobin from human whole blood samples, demonstrating its great potential for practical biological applications” is not absolutely true.

In addition, it is known that in large-scale process hemoglobin is isolated not from whole blood but from erythrocytes previously separated from plasma proteins and leukocytes by centrifugation of cell filtration.

Author Response

In the article “Cu(II)-loaded polydopamine-coated urchin-like titanate micro-spheres as a high-performance IMAC adsorbent for hemoglobin separation” the authors first showed physicochemical characteristics of new IMAC adsorber Cu-PDA-UTMS, and then describe its hemoglobin binding properties. The authors stated that adsorption capacity towards hemoglobin is up to 2.7046 g/L and that 0.1% CTAB elutes 86.8% of adsorbed hemoglobin. They also showed that the pH but not ionic strength influence hemoglobin adsorption, and the elution efficiency depend on stripping reagents. The authors also claimed that Cu-PDA-UTMS shows good reusability up to 6 cycles. However, the results also showed that Cu-PDA-UTMS is not a selective adsorber, specific only for hemoglobin. They adsorbed other proteins, and adsorption efficacy for IgG and BSA is approximately 50% and 20%. The elution efficiency for BSA and IgG is not reported.

Answer: The adsorption efficiency of the adsorbent Cu-PDA-UTMS was up to 92.8% for hemoglobin and less than 20% for BSA. The content of IgG (6-16g/L) in human blood is much less than that of hemoglobin (110-160g/L). The elution efficiency for BSA and IgG is not a critical factor in the isolation of hemoglobin from human blood samples. Therefore, the adsorbent can obviously enrich the target hemoglobin in the presence of the most important interfering protein (BSA). From SDS-PAGE analysis, it can be seen that the proportion of hemoglobin enriched by Cu-PDA-UTMS from human whole blood is significantly higher than that of human whole blood samples before enrichment, which further proves the ability of adsorbent to successfully separate and enrich hemoglobin from complex practical samples.

General comments

Although hemoglobin binding properties of Cu-PDA-UTMS adsorber are promising, I think that the adsorbing characteristics of this matrices have to be studied more profoundly.

I suggest rejection or serious revision which should include new experiments.

Specific comments

Introduction

In Introduction section first paragraph should be deleted and replaced with new one. The sentence “Therefore, the separation and enrichment of hemoglobin from complex biological samples is critical for the treatment and diagnosis of some diseases.” is not clear. Modern diagnostic methods for detecting different hemoglobin opathies do not require neither purified hemoglobin nor large amounts of hemoglobin. Also, is not clear what kind of disease treatments the authors are referring to.

Answer: The sentence “Therefore, the separation and enrichment of hemoglobin from complex biological samples is critical for the treatment and diagnosis of some diseases.” has been be modified in the revised manuscript.

For thar reason I suggest that authors in the first paragraph of Introduction section report the data, existing in scientific literature, on potential biomedical and biotechnological application of purified or partially purified hemoglobin. In the light of existing data, the authors should more precisely explain the aim of this study and to assume potential application of Cu-PDA-UTMS adsorber.

Answer: More details about precisely explain the aim of this study and to assume potential application of Cu-PDA-UTMS adsorber has been added in Introduction section in the revised manuscript.

 Experimental section

The Molecules journal articles have “Materials & Methods” but not “Experimental section”. In addition, this section must be placed after Result and Discussion section. The authors should rearrange the manuscript according the journal demands.

Answer: “Experimental section” has been modified to "Materials and Methods" and "Materials and Methods" section has been adjusted after “Results and Discussion“ section in the revised manuscript.

Subsection 2.1. Chemicals and reagents

The last sentence in subsection “Human whole blood was provided by a healthy volunteer from Shandong Second Medical University” should be followed by the statement which is now in supplementary material “All experiments were performed in compliance with the Helsinki Declaration of 1975 (revised in 2008), as approved by the institutional committee of Shandong Second Medical University on human experiments, and approved by the Ethics Committee of Academic Advisory Board, Shandong Second Medical University, China. Informed consent was obtained from all subjects for being included in this Study”.

In addition, Molecules journal demands that a statement including the project identification code, date of approval, and name of the ethics committee or institutional review board be stated in Section ‘Institutional Review Board Statement’ at the end the article. Therefore, I ask the authors to include this information (first of all, the number and date of issuing the ethical approval) in the “Institutional Review Board Statement”.

Answer: The supplementary material “All experiments were performed in compliance with the Helsinki Declaration of 1975 (revised in 2008), as approved by the institutional committee of Shandong Second Medical University on human experiments, and approved by the Ethics Committee of Academic Advisory Board, Shandong Second Medical University, China. Informed consent was obtained from all subjects for being included in this Study” have been added in the Chemicals and reagents. The number and date of issuing the ethical approval have been added in Institutional Review Board Statement in the revised manuscript.

Subsection 2.4. Protein adsorption behavior

The detailed description of protein adsorption behavior must be given in the main text and not in Supplementary material.

Answer: A detailed description of the adsorption experiments has been added to the main text section in the revised manuscript.

The authors wrote that they mixed 0.1 mg adsorbent and 0.1 mg protein (if the protein concentration was 100 mg/L) in total volume of 1 ml, and then they incubated the suspension for 60 min. However, it is not clear if the adsorbent is washed after this incubation period (to remove unbound proteins and an excess of binding buffer) and if the eluent volume was also 1 ml.

If it's not a case, mass of proteins in starting solution and in the eluent should be calculated and “C” i.e., “protein concentration” should be replaced with the “protein mass” in Formula 1, 2 and 3.

Answer: The sorbent was not washed after incubation and the eluted liquid accumulated into 1ml.

Also, the authors should explain in detail how adsorption performance of the adsorbents are determined with UV-vis spectrophotometer. 

Answer: The detail how adsorption performance of the adsorbents is determined with UV-vis spectrophotometer have been added in the revised manuscript.

Additional remark: The authors did not give the composition of binding buffer(s). They just stated the buffer pH value.

Answer: The binding buffer is 4.0 mmol L-1 Britton-Robinson (BR) buffer and have been added in the revised manuscript.

Results and Discussion

Major comment: The obtained results were not discussed at all, neither in the light of the results of other research groups studying IMAC nor in the light of possible application.

Answer: The obtained results, such as adsorbent adsorption capacity, have been compared with the various other adsorbents, and the specific adsorbent application has been applied to the separation of hemoglobin from human whole blood samples.

Subsection 3.2. Protein adsorption behaviors

  1. Although it is known that any IMAC adsorbent is not selective for a single protein species, and that, because of this properties, Cu-PDA-UTMS also binds other proteins besides hemoglobin, the authors demonstrated preferential binding of hemoglobin only by SDS PAGE of whole blood.

Answer: The IMAC adsorbent has an adsorption efficiency of up to 92.8% for the target hemoglobin and less than 20% for the interfering BSA. SDS-PAGE as an important method for verifying the separation and enrichment of hemoglobin, it can be seen that the proportion of hemoglobin enriched from blood is significantly higher than that of human whole blood samples before enrichment from the SDS-PAGE analysis.

  1. The authors did not explain how they papered whole blood (which normally contains plasma proteins and intact erythrocytes, leukocytes and platelets) to be adsorbed to Cu-PDA-UTMS. If they added whole blood to adsorber, and then isolated hemoglobin, this means that adsorber provoked erythrocyte destruction (lysis) and release of free hemoglobin which than became available for the binding to adsorber.

If this is a case, the author should analyze the binding of erythrocyte cytoskeletal proteins (which can be regarded as unwanted contaminates in some of hemoglobin preparation) to Cu-PDA-UTMS.

Answer: The adsorbent is added to a sample of human whole blood without other processing. A lot of research shows the key factor in the separation of hemoglobin in the human whole blood is the high content BSA interference (ACS Applied Materials & Interfaces, 2017, 9(34): 28273-28280; TrAC Trends in Analytical Chemistry, 2019, 120: 115650). The overall content of erythrocyte cytoskeletal proteins is less than the amount of hemoglobin. Thus the binding of erythrocyte cytoskeletal proteins is not the main factor for the separation of hemoglobin in the human whole blood.

  1. The author did not give the characteristics of SDS PAGE. They only reported the name of electrophoresis devices) but no other important data: method for preparing the samples for SDS PAGE, used sample buffer, gel thickens and percentage, protein load, name and manufacturer of protein makers. From my point of view the PAGE is not well performed and bands are not clear. Also, the authors should harmonize the marking of gel lines in the figure (lanes labeled with number) and in the text (lines labeled with letters).

Answer: Method for preparing the samples for SDS-PAGE, used sample buffer, gel thickens and percentage, protein load, name and manufacturer of protein makers have been added in the in the revised manuscript. From SDS-PAGE analysis, it can be seen that the proportion of hemoglobin enriched from blood is significantly higher than that of human whole blood samples before enrichment. We have been unified the description the marking of gel lines by using with number.

  1. Molecular mass of hemoglobin is not 14.3 kD but 64.5 kDa. The mass of 14.3 kDa correspond to mass of alpha or beta globin chains. During SDS PAGE hemoglobin dissociate to their monomers, i.e., alpha and beta globin chains.

Answer: The mass of 14.3 kDa have been changed to the α or β globin chains of Hb in the revised manuscript.

  1. The authors did not analyze if eluted hemoglobin is in its native Fe2+ tetramer conformation or denatured as a consequence of interaction with matrices. To verified it, more method is necessary to be applied and SAD-PAGE is not valid for such analysis. Without data on conformational stability of isolated hemoglobin the reported results can be regarded as preliminary.

Answer: Experiments by using circular dichroism to test the structure of hemoglobin after adsorption and elution by adsorbents have been added in Figure S4 in the revised manuscript.

  1. The authors did not analyze if lipids or lipoproteins (from whole blood i.e., from plasma and erythrocyte membrane) nonspecifically interacted with Cu-PDA-UTMS.

Answer: Since human whole blood is a complex biological sample, the adsorbent has good adsorption selectivity for target proteins in the presence of important interfering proteins. Affected by the actual conditions, it is not feasible to analyze the selectivity of many other proteins with small content in human whole blood one by one.

  1. The authors used whole blood which normally contains leucocytes. Leukocytes contain nucleic acids which can be released during the procedure and contaminate the isolated hemoglobin. This means that DNA and RNA binding to Cu-PDA-UTMS should be analyzed.

Answer: Due to the influence of experimental conditions and techniques, it is difficult to study the relevant conditions of DNA and RNA adsorption in human whole blood. In addition, it can be seen from the SDS-PAGE that the separation and purification effect of the adsorbent when used to isolate hemoglobin from only the unprocessed whole blood in the case of many interfering substances has achieved satisfactory results.

Conclusions

I think that statement “Cu-PDA-UTMS has successfully isolated hemoglobin from human whole blood samples, demonstrating its great potential for practical biological applications” is not absolutely true.

In addition, it is known that in large-scale process hemoglobin is isolated not from whole blood but from erythrocytes previously separated from plasma proteins and leukocytes by centrifugation of cell filtration.

Answer: Because hemoglobin is in blood cells and blood cells are in whole human blood. In this experiment, the purpose of direct separation and enrichment of hemoglobin in untreated human whole blood can be achieved through the direct absorption of the adsorbent. Many previous studies have also proved that this kind of solid phase extraction method is feasible (Talanta, 2019, 200: 100-106; Journal of Solid State Chemistry, 2017, 253: 219-226; ACS omega, 2022, 7(26): 22633-22638; Artificial Cells, Nanomedicine, and Biotechnology, 2017, 45(1): 39-45).

Reviewer 2 Report

Comments and Suggestions for Authors

The present study titled " Cu(II)-loaded polydopamine-coated urchin-like titanate micro-spheres as a high-performance IMAC adsorbent for hemoglobin separation" by Zhang et al. is particularly interesting. Researchers prepared a novel IMAC adsorbent polydopamine-coated urchin-like titanate microspheres (Cu-PDA-UTMS), and they used for the separation of hemoglobin protein. Thus, they showed that Cu-PDA-UTMS exhibit favorable selectivity for hemoglobin adsorption and have a desirable adsorption capacity towards hemoglobin. Thus, they resulted, this high-performance IMAC adsorbent provides a new way for the effective separation and purification of hemoglobin from complex biological samples.

The quality of the writing is good, and the general structure of the manuscript is well-structured. However, there are a few comments to improve the manuscript:

-The quality of Scheme 1 and figure 3-5 should be increased.

- Full-length figure legend will be helpful to understand the figures more deeply.

- The content of the introduction section should be extended by adding more information about other separation techniques of hemoglobin and their limitations and strengths. Please compare and discuss features of separation techniques.

-Please summarize method of “Protein adsorption behavior” in the manuscript file.

- In the results section, the data obtained should be written more formally without saying “We investigated”

- The conclusion section should be added to highlight future remarks, importance, and relevance of their work.

Author Response

The present study titled " Cu(II)-loaded polydopamine-coated urchin-like titanate micro-spheres as a high-performance IMAC adsorbent for hemoglobin separation" by Zhang et al. is particularly interesting. Researchers prepared a novel IMAC adsorbent polydopamine-coated urchin-like titanate microspheres (Cu-PDA-UTMS), and they used for the separation of hemoglobin protein. Thus, they showed that Cu-PDA-UTMS exhibit favorable selectivity for hemoglobin adsorption and have a desirable adsorption capacity towards hemoglobin. Thus, they resulted, this high-performance IMAC adsorbent provides a new way for the effective separation and purification of hemoglobin from complex biological samples.

The quality of the writing is good, and the general structure of the manuscript is well-structured. However, there are a few comments to improve the manuscript:

Comment 1. The quality of Scheme 1 and figure 3-5 should be increased.

Answer: The quality of Scheme 1 and Figure 3-5 has been increased in the revised manuscript.

Comment 2. Full-length figure legend will be helpful to understand the figures more deeply.

Answer: Full-length figure legend have been modified in the revised manuscript.

Comment 3. The content of the introduction section should be extended by adding more information about other separation techniques of hemoglobin and their limitations and strengths. Please compare and discuss features of separation techniques.

Answer: More information about features of separation techniques of hemoglobin have been added in the revised manuscript.

Comment 4. Please summarize method of “Protein adsorption behavior” in the manuscript file.

Answer: The method of “Protein adsorption behavior” has been added in the revised manuscript.

Comment 5. In the results section, the data obtained should be written more formally without saying “We investigated”

Answer: “We investigated” has been deleted in the revised manuscript.

Comment 6. The conclusion section should be added to highlight future remarks, importance, and relevance of their work.

Answer: The highlight future remarks, importance, and relevance of their work have been added in the revised manuscript

Reviewer 3 Report

Comments and Suggestions for Authors

Reviewers Comments

The authors have submitted a nicely written and well-presented research article and surely would be of interest to scientists in material synthesis and adsorption. However, incorporating the following suggestions and comments definitely improve the quality of the submitted manuscript:

1.      The last paragraph of the introduction section usually focuses on the hypothesis and plan of the investigation and not on the results and findings. Please corrected it if possible.

2.      Hydrochloric acid is mentioned twice in the Chemical and Reagents section.

3.      What is meant by renewed in line 111?

4.      The word “violently stirred” in line 124 needs rewording.

5.      Typo error in line 187, From Figure 3, he success.

6.      Figure 3 d – N2 adsorption/desorption – in the figure it says absorption instead of adsorption.

7.      Line 234, “we then studied”. The sentence needs rephrasing.

8.      Sentence beginning inline 238 and ending in 240 is a bit confusing and needs to be rephrased.

9.      How did you determine the adsorption capacity of 2704.6 from Figure 5a? Please give more details.

10.   You mentioned adsorption capacity in line 279 but in the table 1, binding capacity is written. Are these two different things?

11.   Acetic acid, Na2CO3 not mentioned in the Chemicals and Reagents section 3.1.

12.   There is mention of ammonia solution in section 3.1 but where it is used in the manuscript is not clear.

13.   In the reusability section, “continuous adsorption/desorption” is not clear. More details should be added in this section.

Comments on the Quality of English Language

minor typo errors are there and should be corrected.

Author Response

The authors have submitted a nicely written and well-presented research article and surely would be of interest to scientists in material synthesis and adsorption. However, incorporating the following suggestions and comments definitely improve the quality of the submitted manuscript:

Comment 1. The last paragraph of the introduction section usually focuses on the hypothesis and plan of the investigation and not on the results and findings. Please corrected it if possible.

Answer: The last part of the introduction was modified in the revised manuscript.

Comment 2. Hydrochloric acid is mentioned twice in the Chemical and Reagents section.

Answer: Redundant hydrochloric acid has been deleted in the revised manuscript.

Comment 3. What is meant by renewed in line 111?

Answer: The word of “renewed” has been replaced by the word of “replaced” in the revised manuscript.

Comment 4. The word “violently stirred” in line 124 needs rewording.

Answer: The word of “violently stirred” has been replaced by the word of “violently stirring” in the revised manuscript.

Comment 5. Typo error in line 187, From Figure 3, he success.

Answer: Typo error in line 187, From Figure 3 has been corrected in the revised manuscript.

Comment 6. Figure 3 d – N2 adsorption/desorption – in the figure it says absorption instead of adsorption.

Answer: The word of “absorption” has been replaced by the word of “adsorption” in the revised manuscript.

Comment 7. Line 234, “we then studied”. The sentence needs rephrasing.

Answer: The sentence has been rephrased in the revised manuscript.

Comment 8. Sentence beginning inline 238 and ending in 240 is a bit confusing and needs to be rephrased.

Answer: The sentence has been rephrased in the revised manuscript.

Comment 9. How did you determine the adsorption capacity of 2704.6 from Figure 5a? Please give more details.

Answer: It is based on Figure 5a, when the protein concentration is 0.7 mg mL-1, the maximum of the column coordinates is the adsorption capacity of 2704.6.

Comment 10. You mentioned adsorption capacity in line 279 but in the table 1, binding capacity is written. Are these two different things?

Answer: Adsorption capacity and binding capacity refer to the same thing. It has been unified to describe the adsorption capacity only in the revised manuscript.

Comment 11. Acetic acid, Na2CO3 not mentioned in the Chemicals and Reagents section 3.1.

Answer: Acetic acid and Na2CO3 has been added in the revised manuscript.

Comment 12. There is mention of ammonia solution in section 3.1 but where it is used in the manuscript is not clear.

Answer: Ammonia solution has been deleted in the revised manuscript.

Comment 13. In the reusability section, “continuous adsorption/desorption” is not clear. More details should be added in this section.

Answer: More details about “continuous adsorption/desorption” in the reusability section has been added in the revised manuscript.

Round 2

Reviewer 1 Report

Comments and Suggestions for Authors

The authors improved the manuscript but some items must be explained more precisely before publishing the work.

My major comment is related to discussion of the results. I have already written that Cu-PDA-UTMS adsorber is promising, but I still think that the question of protein, lipid and nucleic acid contaminations is very important, and should be discussed. It is well known that this contaminant, especially cytoskeletal proteins (even in trace amount) might produce harmful side effects if they are released (in case of pathological conditions) or found (as a part of a therapeutic) in the blood.

Because of that the authors  should note in Discussion or Conclusion that the hemoglobin purification/separation is multi-stage process where the number of purification steps depends on application of the isolated hemoglobin (diagnostic reagent, food additive, cell culture additive, therapeutic agent or it just needs to be removed due to interference with the molecules to be analyzed). In light of this new adsorber, with proved good characteristics, can be used as one step in complex hemoglobin purification process.

Minor comment is related to UV-Vis spectrometry. The authors should give the wavelength range, or individual wavelengths used in the hemoglobin analysis.

My criticism related to SDS PAGE remain unchanged. This method is used as a rough estimate of the degree of purity of a protein. Also, with applied Coomassie blue staining is not possible to analyze nucleic acids in a very complex, protein/lipoprotein/lipid mixture such as peripheral blood. However, the authors improved the manuscript by CD analysis showing almost preserved alpha helix structure. But if the authors continue their work with hemoglobin additional  sophisticated methods will be necessary to prove preserved structure and function of isolated hemoglobin.

All my comments could be corrected relatively easily. The paper can be published in Molecule journal without sending me for additional review if the author made all suggested corrections .

Author Response

The authors improved the manuscript but some items must be explained more precisely before publishing the work.

My major comment is related to discussion of the results. I have already written that Cu-PDA-UTMS adsorber is promising, but I still think that the question of protein, lipid and nucleic acid contaminations is very important, and should be discussed. It is well known that this contaminant, especially cytoskeletal proteins (even in trace amount) might produce harmful side effects if they are released (in case of pathological conditions) or found (as a part of a therapeutic) in the blood.

Because of that the authors should note in Discussion or Conclusion that the hemoglobin purification/separation is multi-stage process where the number of purification steps depends on application of the isolated hemoglobin (diagnostic reagent, food additive, cell culture additive, therapeutic agent or it just needs to be removed due to interference with the molecules to be analyzed). In light of this new adsorber, with proved good characteristics, can be used as one step in complex hemoglobin purification process.

Answer: The details about “the hemoglobin purification/separation is multi-stage process where the number of purification steps depends on application of the isolated hemoglobin (diagnostic reagent, food additive, cell culture additive, therapeutic agent or it just needs to be removed due to interference with the molecules to be analyzed). In light of this new adsorber, with proved good characteristics, can be used as one step in complex hemoglobin purification process.” has been added in the revised manuscript.

Minor comment is related to UV-Vis spectrometry. The authors should give the wavelength range, or individual wavelengths used in the hemoglobin analysis.

Answer: The individual wavelength used in the hemoglobin analysis has been added in the revised manuscript.

My criticism related to SDS PAGE remain unchanged. This method is used as a rough estimate of the degree of purity of a protein. Also, with applied Coomassie blue staining is not possible to analyze nucleic acids in a very complex, protein/lipoprotein/lipid mixture such as peripheral blood. However, the authors improved the manuscript by CD analysis showing almost preserved alpha helix structure. But if the authors continue their work with hemoglobin additional sophisticated methods will be necessary to prove preserved structure and function of isolated hemoglobin.

Answer: Thank you very much for the reviewer's advice. In subsequent studies, hemoglobin additional sophisticated methods would be explored to prove preserved structure and function of isolated hemoglobin.

All my comments could be corrected relatively easily. The paper can be published in Molecule journal without sending me for additional review if the author made all suggested corrections.